# Cerebral Vasculopathy in Children with Neurofibromatosis Type 1

**DOI:** 10.3390/cancers15205111

**Published:** 2023-10-23

**Authors:** Laura L. Lehman, Nicole J. Ullrich

**Affiliations:** Department of Neurology, Boston Children’s Hospital and Harvard Medical School, Boston, MA 02115, USA

**Keywords:** neurofibromatosis type 1, children, cerebral vasculopathy, moyamoya syndrome, stroke

## Abstract

**Simple Summary:**

Neurofibromatosis type 1 is a common autosomal dominant inherited condition, with variable phenotype among affected individuals. Abnormalities of the cerebral vasculature are an acknowledged but poorly understood complication, leading to high morbidity and mortality for individuals with neurofibromatosis type 1. The current review discusses the different types of cerebral vasculopathy, the clinical presentation and diagnosis, followed by proposed pathogenesis. Medical and surgical management options are presented, with a goal to prevent lasting neurologic injury.

**Abstract:**

Cerebrovascular abnormalities are a severe and often underrecognized complication of childhood neurofibromatosis type 1 (NF1). There are no prospective studies of cerebral vasculopathy in NF1; thus, the estimated frequency of vasculopathy varies between studies. The data is difficult to interpret due to the retrospective data collection and variability in whether imaging is done based on screening/surveillance or due to acute neurologic symptoms. The prevalent NF1-associated cerebral vasculopathy is moyamoya syndrome (MMS). Vascular changes can present without symptoms or with acute TIA or stroke-like symptoms or a range of progressive neurologic deficits. Advanced imaging may enhance sensitivity of neuroimaging in children. Medical and/or surgical interventions may prevent short- and long-term complications. Challenges for establishment of a screening protocol for cerebral vasculopathy in children with NF1 include the relatively large number of patients with NF1, the potential need for sedation to achieve quality imaging and the broad age range at time of detection for cerebral vascular changes. The goal of this review is to present the epidemiology, clinical presentation, imaging features and medical/surgical management of cerebral arteriopathies in children with NF1.

## 1. Introduction

Neurofibromatosis type 1 (NF1) is an autosomal dominant, multisystem disorder with an incidence of approximately 1:3000 individuals [1]. NF1 is a common condition, characterized by variable phenotype, lifelong risk for malignancy and high risk for morbidity. The diagnostic criteria for NF1 have been recently been updated by an international consensus panel [2]. Cerebral vasculopathy is a well-recognized but poorly understood and serious complication of NF1; blood vessel abnormalities involving the central nervous system can lead to high morbidity and mortality in this patient population. The goals of this review are: to describe the spectrum of cerebral vasculopathy in pediatric NF1; to discuss the pathogenesis of cerebrovascular changes in NF1; and, to review the clinical presentation and management of cerebral vasculopathy in NF1. Greater understanding about the relationship of NF1 to the underlying vascular changes can lead to new approaches to both treat and, ultimately, prevent cerebrovascular events for individuals with NF1.

## 2. Pathogenesis

Cerebrovascular disease (CVD) refers broadly to a group of conditions that affect the blood vessels and blood supply to the brain. Several studies have examined the prevalence of CVD in children with NF1 (see Table 1 for prevalence rates); however, the true incidence of CVD in NF1 is difficult to ascertain, as not all individuals with NF1 undergo routine/surveillance neuroimaging and/or neurovascular imaging.

Cerebral arterial abnormalities found in patients with NF1 have been described along the entire arterial tree. In the brain, the most commonly recognized abnormalities include moyamoya syndrome, stenosis, occlusion, hypoplasia, fusiform dilation, ectasia, aneurysm and arteriovenous fistulae (AVF). Estimates of prevalence rate in patients with NF1 range from 2.3 to 7.4% ascertained from retrospective cohort studies and case series [3,4,5,6,7,8,9,10,11,12]. Cerebral aneurysms have been reported in association with NF1 in cohort studies and case series [10,11]; however, a retrospective autopsy study did not find any association between NF1 and cerebral aneurysms [13].

There is a predilection for the anterior circulation vessels, thought because the internal carotid arteries and branches are derived embryologically from the neural crest cells, whereas the posterior circulation arises from mesoderm [14].

Retrospective cohorts and case series have also assessed the relationship between NF1 and CVD with regards to age at diagnosis of NF1, age at time of diagnosis of CVD, type of CVD, and clinical manifestations (Table 1). Differences in the demographics, overall incidence, mean age at presentation and types of cerebral vasculopathy between studies is likely contributed by the retrospective nature of the case series and that in some cohorts, MRI is obtained only because of new neurologic symptoms and not in all patients while other studies represent cohorts wherein all patients underwent surveillance neuroimaging. Recent published management guidelines do not recommend routine MRI screening for optic pathway glioma [15,16].

**Table 1 cancers-15-05111-t001:** Summary of case series reporting cerebrovascular changes associated with NF1.

	Type of Study	N	NF1 Age at Diagnosis (Years)	CVD Age at Diagnosis (Years)	Imaging Modality	Type of CVD	Vessels Involved	Reason for Imaging	Signs/Symptoms (N)	Treatment (N)
[11]	Case series	N = 353 MRI 316 CVD 8	1.4	7.3	MRI MRA DSA	Stenosis/Occlusion Ischemia Ectasia Moyamoya Aneurysm Hypoplasia	ICA MCA PCA	Screening	Hemiparesis Seizure Asymptomatic (7)	Revascularization (3)
[4]	Case series	N = 698 MRI 144 CVD 7	1.5–5	6.8	MRI MRA DSA	Hypoplasia Stenosis/Occlusion Collateral vessels	ICA ACA MCA PCA	Clinical indication	Seizure (1) Hemiparesis (1) Paresthesia (1) TIA (1) Asymptomatic (5)	Revascularization (1)
[10]	Retrospective Cohort	N = 419 MRI 266 CVD 17	2	5.2	MRI MRA CT CTA DSA	Stenosis/Occlusion Aneurysm Moyamoya	ICA ACA MCA	Clinical indication	Seizure (3) Hemiparesis (1) Speech delay (2) Bell’s palsy (1) LD (4) Paresthesias (1) Weakness (2) Hyperreflexia (2) ADD (1) Hyperphagia (1) Dysphasia (1) Infantile Spasms (1) Hypertonia (1) Clonus (1) Ptosis (1) RAPD (1) Hemiballisumus (1) Dystonia (1) Aphasia (1) Fine motor delay (1)	Revascularization (6)
[7]	Case series	N = 398 MRI 312 MRA 143 CVD 15	4.3	11.7	MRI MRA CTA DSA	Stenosis/Occlusion Ischemia Moyamoya	ICA MCA PCA VA	Screening OR symptoms	Headache (5) Seizures (2) Asymptomatic (7)	Revascularization (1)
[9]	Cross-sectional	N = 181 MRI 77 CVD 12	8	3–13	MRI MRA DSA	Moyamoya Stenosis/Occlusion Tortuosity Elongation Displacement DVA	ACA MCA < ICA PCA VA		Weakness (1) Decreased responsiveness (1) Asymptomatic (10)	Revascularization (2)
[17]	Case series	N = 6	2.7	11.4	MRI MRA DSA SPECT	Moyamoya (6) Stenosis/Occlusion ICH	ICA ACA MCA		TIA (3) Headache (1) ICH (1) Ischemia (1)	Revascularization (5)
[18]	Retrospective Cohort	N = 18	2.9	7.4	MRI MRA DSA	Moyamoya (18)	ACA MCA PCA	Screening (8) Symptoms (10)	Hemiparesis (2) Seizure (2) Headache (6) Asymptomatic (8)	Revascularization (11)
[3]	Retrospective Cohort	N = 24	N/A	7	MRI MRA DSA	Moyamoya (19) Stenosis (5) Aneurysm (1)	ICA MCA	Screening OR symptoms	Hemiparesis (6) TIA/Stroke (6) Seizure (2) Headache (3) Optic pathway glioma (6) Asymptomatic (8)	Revascularization (19)

CVD = Cerebrovascular disorders, MRI = magnetic resonance imaging, MRA = magnetic resonance angiography, DSA = digital subtraction angiography, CT = computed tomography, CTA = computed tomography angiography, SPECT = single-photon emission computerized tomography; ICA = Internal carotid artery; ACA = anterior carotid artery; MCA = middle cerebral artery; PCA = posterior cerebral artery; TIA = transient ischemic attack.

There is lack of consensus as to the association of CVD and causality due to other manifestations of NF1. Moyamoya can occur in individuals with or without NF1, although there is clearly an increased risk to those with NF1 [3,14,19,20]. Moyamoya is also the most frequently observed cerebrovascular abnormality in NF1 and is characterized by progressive stenosis of the internal carotid artery as well as stenosis of the other major intracranial arteries affecting anterior greater than posterior circulation. Moyamoya can be found in patients with NF1 who are asymptomatic when neuroimaging is performed for another indication, as noted above.

Proposed risk factors to develop moyamoya in the setting of NF1 include younger age, known optic pathway glioma and cranial irradiation. In one large, clinic-based, single-institutional study, sex, presence of hypertension and presence of other NF1 features such as plexiform neurofibroma, optic pathway glioma, and focal areas of abnormal signal intensity, were not predictive of nor associated with the presence of vasculopathy of any type [9]. In a separate single institutional case series by Rea et al. [10], optic pathway gliomas were detected in 76% of children with arteriopathy and there was no association with hypertension. Ghosh et al. [7] similarly reported that optic pathway gliomas on imaging were present in 50% of patients with cerebral arteriopathy. Lastly, in a recent large cohort of children with NF1, 13 of 15 (87%) who had undergone surveillance neuroimaging specifically for screening of optic pathway glioma had moyamoya; vascular changes were found on initial imaging in 31% [3].

Children with NF1 may undergo more frequent imaging due to screening for and follow-up of optic pathway glioma, which in turn may lead to detection of otherwise asymptomatic vascular changes and/or an earlier diagnosis of CVD. Despite this association, it is not clear if this the relationship is causal or if it is unrelated and is confounded by the increased frequency of imaging [3,9]. Mechanistically, it has been postulated that the presence of a tumor may release growth factors, which in turn impact the vascular endothelial proliferation [21]. In some circumstances, there may be some compression-related changes to the blood vessels caused by adjacent tumor. And lastly, genetic changes of the tumor and surrounding cells may facilitate vascular changes and subsequent changes to the tumor microenvironment.

Cranial irradiation increases the risk to develop moyamoya in children with NF1. In a single-center retrospective cohort of nearly 450 children treated with cranial irradiation, Ullrich et al. found an increased susceptibility to develop moyamoya in children with NF1; moreover, children with NF1 had a 48% rate of progression of their disease and 56% had radiographic infarct on MRI at diagnosis [22]. Most NF specialists would recommend avoiding radiation therapy in the child with NF1 for a myriad of reasons including: risk for development of vasculopathy, risk of new tumor formation, risk for malignant transformation of existing tumors and increased risk for cognitive changes.

## 3. Associations between Cerebral and Peripheral Vasculopathy

Some individuals with NF1-associated CVD have concurrent stenoocclusive lesions in multiple sites, including cerebral, renal, mesenteric and cardiac vessels. Hypertension is common for individuals with NF1, both with and without moyamoya. In a single-institution retrospective study of 101 pediatric patients with moyamoya disease and available abdominal/renal angiography, 6/101 had hypertension and 8/101 had concurrent renal artery stenosis [23]. A separate study reported an estimated frequency of renal artery stenosis in patients with moyamoya disease as high as 5% [24]. Though controversial, many NF-specialists recommend if a patient has vascular changes in one system, they should also be evaluated for other sites of stenosis/occlusion. Due to the increased risk of hypertension and younger age of onset compared to the general population, children with NF1 should have regular monitoring of blood pressure at medical visits [16]. Persistent elevated blood pressure deserves further evaluation with renal ultrasound and consideration of serum catecholamines, as approximately 1% of individuals with NF1 have underlying renal artery dysplasia/stenosis and another 1% have an underlying pheochromocytoma [25].

## 4. Clinical Symptoms

Age at diagnosis varies in the available retrospective cohort studies and case series. Because the symptoms can be clinically subtle, in some cases just presenting with headache, there is the potential for delayed diagnosis. Moyamoya presents most commonly with stroke but can also present with transient ischemic attacks (TIA), headache, seizures and movement abnormalities such as chorea [26]. These symptoms are often presenting signs of ischemia to a specific region and may be the presenting sign of a TIA or stroke. Gradually over time, the neurologic deficits may become permanent. Adults are more likely to present with hemorrhagic stroke (prevalence 10–40%) [27]. Many individuals with NF1-associated vasculopathy do not have specific symptoms at diagnosis and remain asymptomatic. The vascular abnormalities in those cases are typically discovered incidentally on scans performed for other reasons (Table 1).

Stroke including both hemorrhagic and ischemic, occur at higher rates in both adults and children with NF1. A study examining the US Nationwide Inpatient Sample comparing adult patients with NF1 to controls found adults with NF1 have 20% higher odds of ischemic stroke and almost 2-fold higher odds of hemorrhagic stroke. The odds of stroke in children with NF1 was also higher, with 8-fold higher odds of hemorrhagic stroke and 3-fold higher odds of ischemic stroke [28].

## 5. Imaging Techniques

Overall, there is a low threshold for vascular imaging in patients with NF1 who present with symptoms suggestive of vasculopathy. Several different modalities of imaging are used for diagnosis and screening of CVD in children with NF1, each with potential advantages and disadvantages (Table 2). Young children with NF1 and older children with NF1 and intellectual disabilities may require sedation for imaging. The risk of sedation in children with NF1 is similar to children without NF1.

Routine surveillance imaging with magnetic resonance imaging (MRI) or magnetic resonance angiography (MRA) of the brain in children without symptoms are not typically recommended [16]. Magnetic resonance imaging (MRI) should be used as first line for evaluation of children with NF1 and suspected CVD as it does not require radiation and will pick up small white matter infarcts not be seen on head computed tomography (CT). The use of Fast MRI protocols with MRA can decrease scanner time which improves the ability to obtain MRI in children who have trouble laying still for longer scans. Specific sequence done commonly with MRI called fluid-attenuated inversion recovery (FLAIR) can also demonstrates early signs of vascular changes. The IVY sign, which is a sign of slow cortical blood flow in children with moyamoya and subsequent reduction of IVY sign in children after cerebral revascularization has been associated with lower rate of post-operative stroke [29]. Magnetic resonance angiography (MRA) is also helpful for demonstrating the stenosis, hypoplasia or dilation of cerebral arteries.

While computed tomography angiography (CTA) has associated radiation exposure, it can in some cases give a better visualization of the cerebral arteries. The gold standard for diagnosis of moyamoya is cerebral catheter angiogram (aka digital subtraction angiography, DSA). DSA is recommended for diagnosis of moyamoya as well as for surgical planning to avoid injuring important collaterals during surgery [30] (Table 1). However, DSA itself is associated with a small risk of peri-procedure stroke (estimated as ~0.7%). Additionally, DSA includes exposure to ionizing radiation and requires sedation. In terms of radiation exposure, it is estimated that use of CT scans in children increases risk of leukemia and brain tumors. As noted previously, a specific concern in patients with NF1 is the development of new tumors and secondary malignant degeneration of existing tumors [31]. Thus, although clinical benefits should outweigh absolute risk, radiation exposure should be minimized as much as feasible.

Transcranial doppler (TCD) is a non-invasive technique using ultrasonography to assess cerebral blood flow. TCD is currently used to evaluate cerebral blood flow and to screen for cerebral vasculopathy in sickle cell anemia, where there is a known increased risk for moyamoya syndrome and cerebral vasculopathy [32]. In a pilot study of 40 children with NF1, TCD was assessed as a possible screening method specifically for identifying cerebral vasculopathy [33]. Children with hemodynamic features suggestive of stenosis on TCD (N = 4) subsequently underwent MRA as confirmation and 3 of 4 had arterial stenosis or occlusion and one a normal MRA. This type of assessment could, in theory, be used for screening evaluation in children with NF1, where MRI/MRA can be then used to confirm the diagnosis. The advantages include the non-invasive technique without the need for sedation. TCD is less useful for small vessel disease or dissection, although these are less commonly observed in NF1.

Other studies are used to examine perfusion due to the cerebral vascular abnormalities including SPECT, cerebrovascular reactivity (CVR), arterial spin label perfusion imaging (ASL) and contrast perfusion studies, specifically magnetic resonance perfusion (MRP) or computed tomography perfusion (CTP). These non-invasive imaging techniques may be a helpful tool to evaluate for hemodynamic insufficiency and, therefore, to determine the need for revascularization. SPECT is considered the gold standard for evaluating perfusion deficits in moyamoya, but has limited institutional availability and also includes the risk of exposure to radioisotope. ASL is an MRI perfusion technique that does not have exposure to contrast or radiation but can overestimate the area of hypoperfusion. MRP has exposure to contrast but no radiation. CTP has a significant radiation exposure as well as exposure to radiation. CVR with breath-hold has been a non-invasive MRI technique without radiation or contrast exposure to measure the vascular reactivity and identify areas of steal which has been associated with higher risk of ischemic injury [34,35,36].

Novel imaging techniques such as diffusion magnetic resonance imaging (dMRI) can be used to screen for white matter injury and may correlate with functional outcomes. In adults with moyamoya, dMRI has shown white matter alterations not seen on conventional imaging which are correlated with cognitive deficits [37]. In children with moyamoya without history of stroke or silent infarcts, dMRI demonstrated significant altered white matter in watershed tracts compared to controls [38]. These results suggest that dMRI may be used to determine when children with asymptomatic moyamoya may benefit from surgical intervention.

## 6. Management of Moyamoya Syndrome in Children with NF1

### 6.1. Medical Management

No proven medical therapy or curative treatment exists for the treatment of NF1-associated vasculopathy and there is no established consensus regarding medical management. The rate of progression is variable and disease progression itself can be slow with rare, intermittent events or with rapid neurologic decline. Development of neurologic signs and symptoms can occur months to years after initial presentation, suggesting there might be a role for prevention/intervention to lower the risk for ischemic changes. Supportive medical management may also reduce the risk of complications.

Most patients with moyamoya and NF1 are started on antiplatelet agents such as aspirin to reduce the risk of ischemic symptoms in moyamoya [39]. There are now effective ways to assess for platelet reaction to antiplatelet medications, such as aspirin and clopidogrel. VerifyNow is a commercially available point of care assay with rapid turnaround time, which uses a whole blood sample to assess platelet reactivity by using light transmittance to measure platelet aggregation [40].

Ischemic symptoms and acute events in patients with moyamoya are most often due to hypoperfusion. Good hydration is also encouraged to help intracerebral blood volume. Dehydration may place children with moyamoya at increased risk for TIA or stroke-like symptoms due to decreased flow through collateral blood vessels, which typically do not autoregulate in the same manner as typical blood vessels. In the acute setting, the goal is to increase blood flow to reduce the likelihood that a transient ischemic event will progress to completed stroke. Hydration, normalization of glucose and electrolytes and treatment of seizures are all important for acute management.

Children with moyamoya may also present with other neurologic symptoms such as headaches and seizures. Symptomatic treatment for these conditions is similar for patients with and without moyamoya. Seizures should be treated with antiseizure medications, as uncontrolled seizures could potentially result in hypoperfusion. Prophylactic treatment of headaches is recommended, if occurring frequently and interfering with quality of life [41], although there are no randomized studies to recommend a specific prevention strategy. Calcium channel blockers may be helpful for headaches or migraines and may reduce the severity and reduce symptoms related to TIAs by improving cerebral blood flow through a vasodilatory effect [42]. Calcium channel blockers have also been used to treat acute cerebral vasoconstriction associated with other cerebral vasculidites [43]. Any child with NF1 with persistent hypertension should have consideration for renal ultrasound or mesenteric artery imaging to screen for renal artery stenosis.

### 6.2. Intervention—Surgical

Revascularization surgery is standard of care for all children with symptomatic moyamoya [39]. Based on a meta-analysis [44], 2/3 of individuals have progression over a 5-year period with no treatment compared to 2.6% after surgical revascularization. Risk of peri-operative stroke is between 4–10% [45]. A modified Delphi approach has identified strategies to reduce stroke risk [46]. Some data suggest early intervention with arterial revascularization and pial synangiosis may reduce stroke risk in children with NF1, if implemented prior to development of symptoms [47]. Long-term studies of pediatric patients treated with surgical revascularization followed into adulthood, there is near elimination of recurrent ischemic stroke; however, the risk of de novo hemorrhage increases over time [48]. Some data suggest that revascularization surgery can also help to relieve not only TIAs but also headaches [41]. The type of revascularization surgery can be either direct or indirect bypass surgery. Direct bypass is a revascularization surgery in which the arterial stenosis is bypassed through connection of the superficial temporal artery to the middle cerebral artery. Indirect bypass involves improving the vascularity around the brain by laying on the brain either the temporal artery or muscle or pia. Surgical treatment and consensus on approach to asymptomatic children with moyamoya remain controversial, specifically in children with NF1. Children with NF1 and asymptomatic moyamoya are at risk for stroke if they do not have surgical revascularization and also at risk for peri-operative stroke during surgery, therefore optimal timing remains unknown.

## 7. Surveillance Imaging and Optimal Monitoring for Cerebrovascular Disease in NF1

The common theme of diagnosis of NF1 prior to diagnosis of CVD has led to the suggestion that early and more frequent imaging studies might detect CVD earlier and possibly could prevent complications. This is the “clinicoradiologic dissociation” with a lag of months to years between detection of vascular changes on imaging to the development of ischemic symptoms or fixed neurologic signs secondary to the vasculopathy. Moreover, there may be radiologic progression in the absence of clinical symptoms.

The systematic screening of children with NF1 for early vascular changes is controversial. In most high-volume NF centers, neuroimaging is recommended with the presence of new neurologic signs or symptoms. A recommendation for universal screening of children with NF1 with brain MRI is in itself problematic for several reasons. First, screening of asymptomatic patients is costly given the high incidence of NF1; second, imaging of young children, many of whom may have coincident learning and developmental issues due to the underlying NF1, is likely to necessitate sedation/anesthesia; third, there is no established regimen for imaging/monitoring or intervention once the vasculopathy has been discovered. MRI may be sufficient to detect underlying vasculopathy, although sensitivity is clearly improved when adding MRA. Newer techniques such as ASL may further increase sensitivity to diagnose vasculopathy and provide information regarding cerebral perfusion in a non-invasive manner and without radiation exposure.

For children diagnosed with vasculopathy, there is no published data; however, yearly follow up MR/MRA is reasonable, or sooner with any new neurologic symptoms.

## 8. Conclusions and Future Directions

Limitations resulting from the retrospective nature of the available literature are noted. Several studies do not have adequate details regarding the specific symptoms at time of presentation, variability in imaging studies utilized, lack of standardized grading systems for CVD in this population, lack of data on chronology and frequency of imaging and indication for imaging studies. Despite these limitations, it is evident that earlier recognition of NF1-CVD and moyamoya is required in order to diagnose and treat with revascularization surgery prior to stroke.

With the advent of more rapid imaging that reduces the total MR scanner time to 5–10 min, more children are likely to be able to undergo imaging without the use of sedation; therefore, the utility of screening and feasibility of screening imaging is improved. Correlative imaging techniques such as perfusion imaging and diffusion MRI as well as TCD may emerge as radiologic biomarkers for non-invasive assessments of arterial stenosis/occlusion in NF1 to identify children at risk. Using dMRI, which can be obtained with clinical imaging, we can identify white matter watershed tracts that are altered. This biomarker could assist with revascularization surgery decision-making in children with NF1 and asymptomatic moyamoya. Given the rarity of cerebrovasculopathy in NF1, prospective, multicentered studies will be required to develop evidence-based guidelines for screening, evaluation, treatment and monitoring of NF1 vasculopathy. This will help to determine potential biomarkers of disease and predictors of progression. Recommendations will need to address the cohort of individuals who are considered clinically asymptomatic and with no prior evidence for infarction/ischemia.

## Figures and Tables

**Table 2 cancers-15-05111-t002:** Imaging modalities–advantages and disadvantages.

	Advantages	Disadvantages
MRI	No radiation Excellent visualization of brain parenchyma Accessible	May miss vasculopathy with no dedicated vascular imaging Requires sedation in children Moderate cost
MRA	No radiation Rapid Accessible	Limited visualization of small blood vessels Requires sedation in children Moderate cost
CTA	Excellent visualization of cerebral arteries Accessible	Poor visualization of brain parenchyma Radiation exposure
DSA	Dynamic view of cerebral vasculature Gold standard	Invasive Requires sedation in children Radiation exposure Small risk of stroke Not universally available in children Most expensive
Transcranial Doppler	Non-invasive Established in other populations with syndromic moyamoya	Lower sensitivity than MRA May be challenging in small children due to patient size and cooperativity Interpretation can be operator dependent Not universally available in children Sensitivity/specificity for NF1-related moyamoya not established Least expensive
ASL	Assessment of perfusion No contrast required No radiation exposure	Can overestimate the area of hypoperfusion Will not detect moyamoya prior to MRI Requires sedation in children Moderate cost
SPECT	Gold standard for perfusion deficits	Radiation exposure Limited institutional availability Requires sedation in children Moderate cost
MR Perfusion	Assessment of perfusion No radiation exposure	Requires contrast Requires sedation in children Moderate cost
CT Perfusion	Assessment of perfusion	Requires contrast Radiation exposure

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
