# Peer review of "Cerebral Vasculopathy in Children with Neurofibromatosis Type 1"

_cancers, 2023, doi:10.3390/cancers15205111_

Round 1

Reviewer 1 Report (New Reviewer)

The manuscript reviews neurofibromatosis type I (NF1) contribution to cerebral vasculopathy, leading to high morbidity and mortality. The article provides a new light on the pathology effects of NF1. Typically, NF1 is associated with brain gliomas. However, this manuscript opens the door to other consequences of NF1. Thus, a gap in NF1 pathogenesis has been well-covered in this review. Much of the data on vasculopathy resulting from NF1 is retrospective. Thus, this review helps consolidate previous findings. Although this review does not contain a methodology section as a research article, this review summarizes in tables various techniques of non-invasive imaging to gauge vasculopathy. Their conclusions are consistent with the summary of reviewed findings with the appropriate references. All the tables are complete, but this review would benefit from a table listing the abbreviations with their definitions fully described. 

Author Response

We appreciate the reviewer's comments and suggestions.

The abbreviations are noted in the manuscript itself; if the editorial staff requests a separate table with abbreviations, we can so provide.  It was not part of the submission as it was not previously requested.

Reviewer 2 Report (New Reviewer)

The authors have undertaken a review of cerebrovascular disease in NF1. This paper appears to have already been through one round of reviews. Overall, it is very well written but the authors have missed the important literature on mortality associated with cerebrovascular disease in NF1. They need a section on this in the paper as this is a potentially unbiased assessment although the big US death certificate study is likely to have missed many NF1 deaths as it required Neurofibromatosis on the certificate. There are however, highly ascertained population based studies in Finland, France and the UK which should be looked at.

Specific comments

1.       ‘Neurofibromatosis type 1 (NF1) is an autosomal dominant, multisystem disorder with an incidence of approximately 1:3000 individuals [1]’ -this reference appears to have been changed. This is not an epidemiological study and does not provide an accurate reflection of birth incidence which is from 1 in 2000 in Finland to 1 in 2500-2,700 in the UK. Please correct with original epi references

2.       ‘Cerebral vasculopathy is a well-recognized but poorly understood and serious complication of NF1; blood vessel abnormalities involving the nervous system can lead to high morbidity and mortality in this patient population.’ -This is unreferenced an the authors fail to refer to mortality papers in NF1 that address cerebrovascular causes as a proportion. This is a major failing of this review see https://pubmed.ncbi.nlm.nih.gov/11283797/ and https://pubmed.ncbi.nlm.nih.gov/21694737/

Author Response

We thank the reviewer for their comments.  

1. Concerns re: reference to epidemiology:  I have chosen the following article, which is a systemic review and meta-analysis and includes both of the articles referenced by the reviewer.

Lee TJ, Chopra M, Kim RH, Parkin PC, Barnett-Tapia C. Incidence and prevalence of neurofibromatosis type 1 and 2: a systematic review and meta-analysis. Orphanet J Rare Dis. 2023 Sep 14;18(1):292. doi: 10.1186/s13023-023-02911-2. PMID: 37710322; PMCID: PMC10500831.

2. Contribution of cerebrovascular disease in NF1 to mortality:  We appreciate the reviewer's comments, however it is quite difficult to ascertain the deaths in these reports specifically to NF1-related cerebrovascular disease, which is why they are not included.  

  • The article by Evans, et al (https://pubmed.ncbi.nlm.nih.gov/21694737/) refers to mortality from tumors of NF1; within this cohort of 1186 persons with NF1, two had death from low grade glioma treated with radiation.  These were attributed to vascular etiology, but given the known mortality of recurrent low grade glioma it is difficult to discern, so this article was not selected.
  • the other recommended article by Rasmussen et al (https://pubmed.ncbi.nlm.nih.gov/11283797/) reports an increase in deaths in the US NF1 population attributable to cerebrovascular disease.  There are several caveats when interpreting the results of this manuscript.  The first is that neurofibromatosis had to be listed as a cause of death.  The second is that the attribution is dependent on ICD9 codes, which have significant limitations.  For example, the codes selected include 438.0 (Cognitive deficits is a medical classification as listed by WHO under the range -CEREBROVASCULAR DISEASE (430-438)).  I would not consider cognitive deficits to be a specific cerebrovascular disease nor a common cause of mortality.  In addition, it included 435.0 (Transient ischemic attack).  By definition, a TIA is time-limited and reversible, so really cannot be a cause of death.  Lastly, it also included 436.0 (Acute, but ill-defined cerebrovascular disease).  If one simply removed the ones labeled as cognitive deficits and TIA, this would make the recorded number not significantly different than the expected number. 
  • We therefore choose not to refer to either article, due to these inherent limitations

Reviewer 3 Report (New Reviewer)

The authors have responded well to all queries raised by both reviewers. 

Author Response

We thank the reviewer and acknowledge no further changes recommended.

This manuscript is a resubmission of an earlier submission. The following is a list of the peer review reports and author responses from that submission.

Round 1

Reviewer 1 Report

The authors present a very interesting review on the cerebral vasculopathies in children with NF1. The review is concise, complete, easy to read and full of practical suggestions some of which are also summarized in the tables.

I have only a few suggestions to clarify some issues that the authors have otherwise already explained:

- the authors should give us also a prospective view of what happens in the adult age. Are there more frequent signs, are there more frequent complications, are there children with long-standing vasculopathy more at risk for severe vascular complications?

- in couple of points the authors state that the absence of RMN screening does not allow a direct estimate of vascular diseases in children. It should be stressed that several latest guidelines argue against a RMN screening for optic glioma (cite for example the ERN Genturis guidelines recenly published on European J of Medical Genetics

-   Finally, I am sure that most of the readers will try to find in the review the frequency of the imaging controls after a vasculopathy has been diagnosed in a child (or in an adult). The authors clearly show there are no data about, but also in the absence, the authors should give at least their point of view.

Reviewer 2 Report

This paper is a review of the epidemiology, clinical presentation, imaging features, and medical/surgical management of cerebral arteriopathies in children with neurofibromatosis type 1 (NF1). Cerebrovascular abnormalities are a severe and often underrecognized complication of NF1, and the prevalent NF1-associated cerebral vasculopathy is moyamoya syndrome (MMS). Advanced imaging may enhance sensitivity of neuroimaging in children, and medical and/or surgical interventions may prevent short- and long-term complications. The paper discusses the challenges in establishing a screening protocol for cerebral vasculopathy in children with NF1. I have reviewed the manuscript and provided comments in the attached document.
